# Impact of Palliative Care in Evaluating and Relieving Symptoms in Patients with Advanced Cancer. Results from the DEMETRA Study

**DOI:** 10.3390/ijerph17228429

**Published:** 2020-11-14

**Authors:** Oscar Corli, Giacomo Pellegrini, Cristina Bosetti, Luca Riva, Matteo Crippa, Emanuele Amodio, Gianlorenzo Scaccabarozzi

**Affiliations:** 1Department of Oncology, Unit of Pain and Palliative Care Research, Istituto di Ricerche Farmacologiche Mario Negri IRCCS, 20156 Milan, Italy; oscar.corli@marionegri.it; 2Fondazione Floriani, Via Privata Nino Bonnet, 2-20154 Milan, Italy; m.crippa@fondazionefloriani.eu; 3Department of Oncology, Laboratory of Methodology for Clinical Research, Istituto di Ricerche Farmacologiche Mario Negri IRCCS, 20156, Milan, Italy; cristina.bosetti@marionegri.it; 4Dipartimento Fragilità/Rete Locale Cure palliative, ASST Lecco, 23900 Lecco, Italy; lu.riva@asst-lecco.it (L.R.); g.scaccabarozzi@asst-lecco.it (G.S.); 5Department of Health Promotion, Mother and Child Care, Internal Medicine and Medical Specialties (PROMISE), University of Palermo, 90127 Palermo, Italy; emanuele.amodio@unipa.it

**Keywords:** cancer patients, palliative care, symptoms

## Abstract

*Background:* Cancer patients experience multiple symptoms throughout the course of the disease. We aimed to provide a comprehensive analysis of the symptom burden in patients with advanced cancer at admission to specialist palliative care (PC) services and seven days later to estimate the immediate impact of PC intervention. *Patient and methods:* The analysis was based on an observational, prospective, multicenter study (named DEMETRA) conducted in Italy on new patients accessing network specialist PC centers during the period May 2017–November 2017. The prevalence and intensity of symptoms were assessed at baseline and after seven days using three tools including the Edmonton Symptom Assessment System (ESAS). *Results:* Five PC centers recruited 865 cancer patients. Thirty-three different symptoms were observed at the baseline, the most frequent being asthenia (84.9%) and poor well-being (71%). The intensity of the most frequent symptoms according to ESAS ranged from 5.5 for asthenia to 3.9 for nausea. The presence and intensity of physical symptoms increased with increasing levels of anxiety and depression. After seven days, prevalence of nausea and breathlessness as well as intensity of almost all symptoms significantly decreased. *Conclusions:* The study confirmed the considerable symptom burden of patients with advanced cancer. PC intervention has significantly reduced the severity of symptoms, despite the patients’ advanced disease and short survival.

## 1. Introduction

Palliative care (PC) is mainly addressed to the evaluation and treatment of physical, functional, psychological, social, and spiritual symptoms throughout the course of chronic and progressive diseases, especially in the advanced and terminal phases [1,2], with the aim of improving, or at least preserving, the patients’ quality of life.

Most of the evidence in recent decades has been related to the treatment of symptoms in cancer patients, at any stage of the disease and of whatever causal origin. In fact, symptoms may derive from the primary disease or its secondary localizations, but also from cancer therapies and even from treatments aimed to control comorbidities or other symptoms such as pain [3,4,5,6,7,8,9]. Nausea and vomiting, for instance, may be due to the localization of the tumor, to impaired gastric emptying or delayed gastrointestinal transit, to antineoplastic agents or radiation therapy, or to opioids given to relieve pain [10,11]. In the patient’s evaluation, the distinction of the symptom’s causal factors may be irrelevant, but for the physician, understanding the pathogenetic mechanisms is fundamental to setting up specific therapies according to the cause. Moreover, the prevalence and severity of symptoms vary across the phases of illness [12]. Pain, nausea, and depression have been reported to be relatively stable over the last six months of life, while dyspnea, drowsiness, poor well-being, lack of appetite, and asthenia tend to increase in severity over time. In particular, the patient’s symptom burden changes during the last days of life, when PC becomes the specific and often only therapeutic approach [13]. The management of this clinical picture—with some symptoms tending to disappear and others taking their place—is a central and crucial issue in controlling patient suffering.

In previous studies, only selected aspects of these symptoms were considered. We therefore carried out this analysis with the objective of providing a comprehensive evaluation of the symptom burden in patients with advanced cancer from the moment of admission to a PC service. We estimated the prevalence and intensity of symptoms, the number of symptoms experienced simultaneously, their prevalence based on the primary tumor site, and the way by which the presence and severity of anxiety and depression could modify the physical symptoms. Symptom prevalence and intensity were reassessed after seven days to estimate the immediate impact of PC, also paying attention to the influence of the care setting on the clinical expression of symptoms.

## 2. Methods

### 2.1. Study Design and Participants

The data included in this analysis were obtained from an observational, prospective, multicenter study (named DEMETRA) aimed at outlining the profile of patients, families, care services, and clinical features from the moment of admission to specialized PC centers. The general characteristics of the DEMETRA study have been published elsewhere [14]. Given the considerable extension of the collected information and the achieved results, the Scientific Committee decided to divide its contents into separate publications, each deepening a specific theme.

Five Italian PC units based in Florence, Forlì, Lecco, Palermo, and Rome participated in the study, which included a total of 1013 individuals suffering from various advanced chronic diseases. The majority of patients were cancer patients (85.4%), treated in three different care settings: home, hospice, and hospital. The present analysis of symptoms focused only on cancer patients given the high prevalence of cancer among the studied population and the specificity of the clinical conditions and symptoms associated with cancer compared to other advanced diseases.

The inclusion criteria for the DEMETRA study were new patients accessing the PC network between May 2017 and November 2017; age ≥ 18 years; presence of a chronic and progressive illness requiring palliative intervention; and written informed consent for study participation and personal data processing. Patients who were already in the care of a PC network and those who could not ensure regular follow-up were excluded.

Eligible patients were recruited over a six-month period (from May to November 2017). Each patient was followed up for 12 months until November 2018.

The present study considered only 865 cancer patients out of 1013 recruited in the DEMETRA study. The flowchart of patients evaluated in the present study is shown in Figure 1. Overall, 780 patients answered the Edmonton Symptom. Assessment (ESAS) questionnaire and were considered for the intensity of symptoms at baseline, while 508 (65.1%) were also evaluated at day 7.

### 2.2. Data Collection

Patient information collected in the DEMETRA study included sociodemographic data, primary tumor site, presence and sites of metastasis, number of concomitant diseases, care setting (hospice, home care or hospital), and symptoms [14]. Data were collected at the baseline visit (day 0) and after seven days (day 7) to assess the first impact of palliative treatments.

Symptoms were assessed using three tools: (1) the interRAI-PC, an instrument evaluating a wide range of PC issues including items related to symptoms [15]; (2) a supplementary list of items to detect symptoms and clinical aspects that were not covered by the interRAI-PC including rattle, dysphagia, dysgeusia, neuropathic pain, bowel occlusion, liver, and kidney functional impairment, paresis, motor disorders, spinal cord compression, pleural effusion and intracranial hypertension; (3) ESAS, which estimates the prevalence of nine main symptoms (pain, asthenia, nausea, depression, anxiety, drowsiness, breathlessness, poor well-being, lack of appetite) and their intensity, measured with a numeric rating scale ranging from 0 to 10 [16,17].

### 2.3. Consent Procedure and Ethical Approval

The DEMETRA study was approved by the Ethics Committee of the A.S.S.T. of Lecco (Lecco, Italy) on 1 December 2016 and subsequently by the institutional review boards of each participating center. Written informed consent for participation in the study and processing of personal data were collected from all recruited patients before any study-related activity was carried out.

### 2.4. Statistical Analysis

Categorical variables were summarized as absolute and relative frequencies, while continuous variables were reported using mean values with the corresponding standard deviation (SD). The differences in the prevalence of symptoms across levels of anxiety and depression were evaluated using Pearson’s chi-squared test, while the Kruskal–Wallis rank-sum test was used to evaluate differences in median symptom intensity between different levels of anxiety and depression. McNemar’s chi-squared test was used to evaluate differences in the prevalence of symptoms between day 0 and day 7. To compare symptom intensity at day 0 and day 7, we applied the paired-samples Wilcoxon test. All data were analyzed using the R software package (version 3.6.1/2019, R Foundation for Statistical Computing, Vienna, Austria).

## 3. Results

The baseline characteristics of patients are summarized in Table 1. Fifty-two percent were women and the mean age was 74 years. Fifty-five percent of patients were taken in charge at home. The most frequent primary tumor sites were the digestive system, lung, urinary, and reproductive system. About 80% of patients had metastases, the most frequent sites being the liver, lung, and bone.

Table 2 lists the 33 symptoms observed at baseline in decreasing order of prevalence. The most frequent symptoms were asthenia, poor well-being, lack of appetite, drowsiness, pain, depression, constipation, anxiety, breathlessness, and insomnia. More than half of the patients had six to twelve simultaneous symptoms and one-tenth had 15 or more symptoms (Appendix A).

The most frequent combinations of concomitant symptoms at baseline are shown in Appendix A. These combinations often included asthenia, lack of appetite, drowsiness, and poor well-being, all of which were present in 46% of patients, while three of these symptoms, variously combined, occurred in over half of the patients. Appendix A shows the prevalence of the main symptoms observed at the baseline according to the primary tumor site. Breathlessness was mainly frequent in lung cancer, pain in pancreatic cancer, anxiety, and drowsiness in breast cancer, and lack of appetite in colorectal cancer.

Nine of the 10 most frequent symptoms were included in the ESAS questionnaire. This allowed us to measure the intensity of these symptoms in the whole population and in symptomatic patients, as shown in Table 3. Patients were defined as symptomatic if a score greater than zero was recorded for that symptom. In the first group, the intensity ranged from 5.2 (asthenia) to 1.7 (nausea); in the second, from 5.5 (asthenia) to 3.9 (nausea).

In Table 4, the intensity of symptoms were compared in patients evaluable at both visits, on day 0 and day 7. After one week, the intensity decreased significantly for all symptoms with the exception of anxiety, depression, and drowsiness. Among patients with severe symptom intensity at baseline (score ≥ 6), we observed a reduction in intensity for all symptoms (Appendix A).

The prevalence and intensity of the four main physical symptoms (pain, breathlessness, nausea, and asthenia) significantly increased with increasing levels of anxiety and depression (Table 5). Anxiety and depression were grouped in asymptomatic patients (intensity = 0), patients with mild to moderate symptoms (score between 1 and 4), and patients with elevated symptoms (score between 5 and 10).

After one week, breathlessness was significantly reduced in patients treated at home, while nausea was significantly reduced in those treated at home or in a hospice (Appendix A). Anxiety and depression significantly increased in patients treated at home and in the hospital, respectively. Pain significantly decreased in all care settings; anxiety only in patients treated in hospital; breathlessness, poor well-being, lack of appetite, nausea, and asthenia in those treated at home; and lack of appetite, nausea, and asthenia in those in hospital (Table 6).

## 4. Discussion

This study evaluated the symptom burden in a cancer patient population during the last days of life, at the time of admission to specialist PC. The decision to appraise only cancer patients was prompted by the need for a clinically homogeneous population, but also by the high proportion of cancer patients among those included in the DEMETRA study. In addition, as already reported [18,19], a cancer patient’s average life expectancy is generally short at the time of admission to PC. The median survival in our population was 29 days, and only about 65.1% of patients were evaluable after the first seven days. Due to the advanced phase of disease, PC health professionals had a short while to reach their therapeutic goals, which were mainly oriented to alleviate symptoms. We sought to evaluate whether and how the symptom profile of patients could be changed in a few days as a result of PC intervention.

We observed the presence of many symptoms at the time of PC admission. Asthenia was the most frequent, followed by poor well-being, lack of appetite, drowsiness, pain, depression, constipation, anxiety, breathlessness, and sleep disorders, which were present in 40% to 70% of patients. The large number of symptoms is characteristic of the advanced stage of cancer, as previously documented [20].

Furthermore, patients experienced a large number of symptoms simultaneously, as also previously observed [9,21]. This is an important and underestimated clinical problem. For patients, the multitude of symptoms is very distressing, while for physicians, treating all the symptoms is complex due to the number of drugs needed, their potential toxicity, and the risk of drug-to-drug interactions. This makes it necessary to decide which symptoms should be primarily treated, with which drugs, and what results are expected. All these latter considerations represent important aspects that open a window to future insights. In our study, frequent combinations of simultaneous symptoms nearly always included asthenia, lack of appetite, drowsiness, and poor well-being, consistent with the findings of others [22,23]. Such combinations entail a negative physical and psychological condition, with lack of energy and interests and detachment from life.

The prevalence of symptoms at baseline differed according to the site of the primary tumor, as also reported elsewhere [24,25]. The association between specific tumor sites and cancer symptoms is clear, even at advanced stages of the disease when widespread metastasis can dilute this specificity. Instead, some symptoms such as pain and asthenia tend to occur in all tumors.

The intensity of symptoms ranged from 5.2 for asthenia to 1.7 for nausea in the overall population, and from 5.5 for asthenia to 3.9 for nausea in symptomatic patients, showing analogies to earlier observations [26,27]. Interestingly, the prevalence and intensity of physical symptoms such as pain, breathlessness, nausea, and asthenia correlated with the presence and severity of anxiety and depression, reflecting the integration between physical and mental suffering.

We also evaluated whether and to what extent PC treatments could change the symptom profile of patients after one week. A week is a short period of time, but not for patients with terminal cancer. Their life expectancy is very limited, and as a consequence, every therapeutic intervention must be immediately effective. The results obtained in one week are therefore an important measure of the clinical impact of PC. In our study, the prevalence of symptoms between day 0 and 7 was unchanged, except for breathlessness and nausea, which tended to decrease significantly. Considering that the clinical situation quickly worsens in advanced cancer, even the containment of a few symptoms should be seen as positive. Conversely, symptom intensity was significantly reduced after one week of treatment, with the exception of depression, anxiety, and drowsiness. In the whole population, the severity of the symptoms significantly decreased on average by about one point (approximatively a 20–25% reduction in symptoms intensity). However, in the group with a severe intensity of symptoms (>6 points), the difference after one week was on average 1.3–1.4 points, similarly close to a 25% reduction in pain intensity. Pain intensity difference was 2.20 points and nausea intensity difference was 2.44, both over 30%. The evaluation as a percentage of pain reduction appears to be more relevant because it relates to the initial intensity value [28]. We believe, therefore, that the reduction in the intensity of symptoms is an important clinical result, indicating the relief of suffering at such a highly critical time.

Finally, changes in the prevalence and intensity of symptoms between days 0 and 7 were evaluated in the three settings of care: home, hospice, and hospital. Some symptoms had a different initial prevalence depending on the setting. Poor well-being, pain, and nausea were more frequent in patients at home, while lack of appetite and depression prevailed in hospice patients. After one week, the reduction of intensity was almost uniform for all symptoms, with negligible differences related to the setting of care.

The main strength of this study is that it provides a comprehensive analysis of various aspects of the symptoms experienced by cancer patients in the final period of life. Its main limitation is the impossibility of describing the treatments given for such symptoms and the subsequent clinical results because the DEMETRA study protocol did not include the collection of these data. It is also important to underline a further limitation. The comparison between day 0 and day 7 was only possible in patients still alive, and therefore we cannot exclude the necessity of considering the possibility of the presence of a selection bias and an overestimation of results.

## 5. Conclusions

The primary aim of this study was to define the clinical symptom profile of 865 patients with advanced cancer from the moment they accessed PC services. The study confirmed that the number and severity of symptoms in these patients was substantial. Multiple symptoms were frequently present and many of them were perceived as severe. Our results make it clear that PC intervention can significantly reduce the severity of symptoms, even at the advanced stages of cancer. This further finding confirms the need for these patients to receive competent PC. In our opinion, such competence concerns not only PC specialists, but also oncologists, general practitioners, and any other physicians dealing with patients at an advanced stage of disease.

## Figures and Tables

**Figure 1 ijerph-17-08429-f001:**
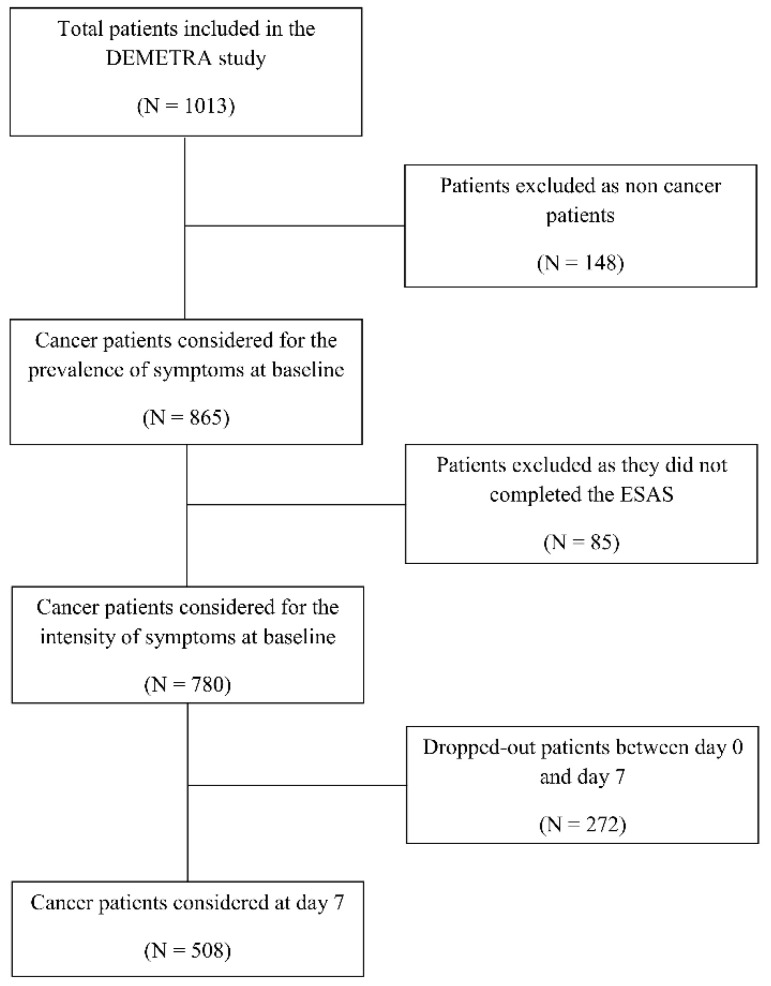
Flowchart of patients included in the study. ESAS, Edmonton Symptom Assessment System.

**Table 1 ijerph-17-08429-t001:** Demographic and clinical characteristics of the 865 study patients at baseline.

Characteristics	Patients *N* (%)
Female	447 (51.7)
Age (years), mean (SD)	74.2 (12.8)
Palliative care centers	
Florence	80 (9.3)
Forlì	160 (18.5)
Lecco	211 (24.4)
Palermo	296 (34.2)
Rome	118 (13.6)
Palliative care setting	
Home	473 (54.7)
Hospital	191 (22.1)
Hospice	201 (23.2)
Primary tumor site	
Lung	181 (20.9)
Digestive system	248 (28.7)
Urinary/reproductive system	167 (19.3)
Head/neck	29 (3.4)
Breast	51 (5.9)
Other	189 (21.9)
Patients with metastasis	683 (79.1)
Site of metastasis	
Liver	298 (43.6)
Lung	277 (40.6)
Bone	227 (33.4)
Brain	96 (14.1)
Other sites	442 (51.1)
Patients with concomitant diseases	486 (56.2)

SD, standard deviation.

**Table 2 ijerph-17-08429-t002:** Prevalence of symptoms in decreasing order of frequency among the 865 patients at baseline.

Symptoms	Patients	
	*N*	%
Asthenia	734	84.9
Poor well-being	614	71.0
Lack of appetite	610	70.5
Drowsiness	543	62.8
Pain	523	60.5
Depression	482	55.7
Constipation	465	53.8
Anxiety	456	52.7
Breathlessness	423	43.5
Insomnia/disturbed sleep	357	41.3
Nausea	348	40.2
Dysgeusia	294	34.0
Dysphagia	277	32.0
Edema	258	29.8
Dry mouth	257	29.7
Bloating/flatulence	188	21.7
Vomiting	176	20.4
Recent falls	160	18.5
Gastroesophageal reflux	151	17.5
Dry cough	137	15.8
Fever	117	13.5
Dizziness/vertigo	80	9.3
Jaundice	80	9.3
Hemorrhage/bleeding	78	9.0
Diarrhea	74	8.6
Profuse sweating	67	7.8
Hallucinations	55	6.4
Hiccups	40	4.6
Fecalomas	36	4.2
Muscle cramps	36	4.2
Death rattle	31	3.6
Seizures	27	3.1
Myoclonus	14	1.6

**Table 3 ijerph-17-08429-t003:** Intensity of the most frequent symptoms in 780 patients at baseline.

Symptoms	All Patients	Symptomatic Patients	
	Mean Intensity (SD) ^a^	*N*	Mean Intensity (SD) ^a^
Asthenia	5.17 (2.53)	734	5.50 (2.25)
Poor well-being	3.98 (2.90)	614	5.05 (2.30)
Lack of appetite	3.93 (3.01)	610	5.03 (2.47)
Drowsiness	3.10 (2.85)	543	4.45 (2.37)
Pain	3.03 (2.78)	523	4.51 (2.19)
Depression	2.68 (2.87)	482	4.33 (2.48)
Anxiety	2.41 (2.70)	456	4.12 (2.33)
Breathlessness	2.10 (2.79)	423	3.43 (3.03)
Nausea	1.75 (2.49)	348	3.91 (2.33)

SD, standard deviation. **^a^** Measured using a numeric rating scale from 0 to 10.

**Table 4 ijerph-17-08429-t004:** Changes in intensity of the main symptoms from day 0 to day 7 in 508 patients.

Symptoms	Intensity, Mean (SD) ^a^			*p* Value ^b^
	Day 0	Day 7	Difference (Day 7–Day 0)	
Asthenia	4.89 (2.59)	4.47 (2.52)	−0.42 (2.37)	<0.001
Poor well-being	3.81 (2.85)	3.4 (2.70)	−0.35 (2.38)	<0.001
Lack of appetite	3.61 (3.03)	3.19 (2.93)	−0.43 (2.63)	<0.001
Drowsiness	3.00 (2.74)	2.77 (2.60)	−0.23 (2.48)	NS
Pain	2.96 (2.66)	2.19 (2.27)	−0.76 (2.15)	<0.001
Depression	2.58 (2.85)	2.60 (2.74)	0.02 (2.33)	NS
Anxiety	2.27 (2.63)	2.27 (2.57)	0.01 (2.34)	NS
Breathlessness	1.92 (2.69)	1.56 (2.49)	−0.36 (1.72)	<0.001
Nausea	1.61 (2.35)	1.28 (2.12)	−0.33 (1.87)	<0.001

NS, not statistically significant. **^a^** Measured using a numeric rating scale from 0 to 10; **^b^**
*p* value for difference between day 7 and day 0.

**Table 5 ijerph-17-08429-t005:** Prevalence and intensity of four physical symptoms in 780 patients according to the intensity of anxiety or depression.

Symptoms		Intensity ^a^ of Anxiety				INTENSITY ^a^ of Depression			
		0	1–4	5–10	*p* Value ^b^	0	1–4	5–10	*p* Value ^b^
		(*N* = 324)	(*N* = 279)	(*N* = 177)		(*N* = 298)	(*N* = 274)	(*N* = 208)	
Asthenia	*N*	294	270	170		269	264	202	
	%	90.7	96.8	96.0	<0.05	90.3	96.4	97.1	<0.01
	Intensity	4.67	5.03	6.32	<0.001	4.34	4.34	6.58	<0.001
Pain	N	198	197	128		188	187	148	
	%	61.1	70.6	72.3	<0.05	63.1%	68.2%	71.2	NS
	Intensity	2.64	3.08	3.64	<0.001	2.79	2.96	3.45	<0.05
Breathlessness	*N*	135	151	90		114	155	107	
	%	41.7	54.1	50.8	<0.05	38.3%	56.6%	51.4	<0.05
	Intensity	1.70	2.30	2.50	<0.01	1.73	2.43	2.19	<0.001
Nausea	*N*	117	137	94		96	137	115	
	%	36.1	49.1	53.1	<0.001	32.2	50.0	55.3	<0.001
	Intensity	1.27	1.69	2.70	<0.001	1.05	1.76	2.72	<0.001

NS, not statistically significant. **^a^** Measured using a numeric rating scale from 0 to 10; **^b^**
*p* value for difference across levels of intensity of anxiety or depression.

**Table 6 ijerph-17-08429-t006:** Changes in intensity of the main symptoms between day 0 and day 7 according to the setting of care in 508 patients overall and in those symptomatic at baseline.

Symptoms	Setting	All Patients			Symptomatic Patients		
		Intensity ^a^, Mean (SD)			Intensity ^a^, Mean (SD)		
		Day 0	Day 7	*p* Value ^b^	Day 0	Day 7	*p* Value ^b^
Asthenia	Home	4.81 (2.59)	4.57 (2.48)	<0.05	5.15 (2.34)	4.80 (2.34)	<0.01
	Hospice	5.12 (2.52)	4.16 (2.45)	<0.01	5.54 (2.12)	4.33 (2.40)	<0.001
	Hospital	5.00 (2.74)	4.25 (2.98)	NS	5.64 (2.18)	4.08 (2.85)	<0.01
	*p* value ^c^	NS	NS		NS	NS	
Poor well-being	Home	3.99 (2.79)	3.70 (2.70)	<0.01	4.92 (2.24)	4.35 (2.46)	<0.001
	Hospice	3.28 (3.01)	2.84 (2.68)	NS	4.79 (2.44)	3.37 (2.68)	<0.001
	Hospital	3.34 (2.88)	2.66 (2.45)	NS	4.74 (2.25)	2.90 (2.45)	<0.001
	*p* value ^c^	<0.05	<0.01		NS	<0.001	
Lack of appetite	Home	3.65 (3.00)	3.30 (2.86)	<0.01	4.85 (2.47)	4.06 (2.70)	<0.001
	Hospice	3.99 (3.27)	3.10 (3.19)	<0.01	5.40 (2.61)	3.68 (3.09)	<0.001
	Hospital	2.50 (2.51)	2.45 (2.86)	NS	4.23 (1.80)	3.31 (2.75)	NS
	*p* value ^c^	<0.05	NS		NS	NS	
Drowsiness	Home	2.98 (2.68)	2.80 (2.55)	NS	4.20 (2.23)	3.57 (2.43)	<0.001
	Hospice	3.00 (2.89)	2.90 (2.74)	NS	4.31 (2.51)	3.69 (2.56)	NS
	Hospital	3.09 (2.96)	2.25 (2.71)	NS	4.69 (2.39)	2.55 (2.85)	<0.01
	*p* value ^c^	NS	NS		NS	NS	
Pain	Home	3.14 (2.63)	2.45 (2.29)	<0.001	4.38 (2.06)	3.15 (2.17)	<0.001
	Hospice	2.08 (2.59)	1.42 (2.04)	<0.01	4.24 (2.11)	2.64 (2.29)	<0.001
	Hospital	3.20 (2.74)	1.66 (2.07)	<0.001	4.55 (2.11)	2.06 (2.22)	<0.001
	*p* value ^c^	<0.001	<0.001		NS	<0.01	
Depression	Home	2.55 (2.82)	2.63 (2.78)	NS	4.24 (2.45)	3.88 (2.60)	<0.05
	Hospice	3.32 (3.00)	2.89 (2.70)	NS	4.55 (2.60)	3.61 (2.64)	<0.05
	Hospital	1.25 (2.25)	1.73 (2.42)	NS	4.23 (2.13)	3.15 (2.54)	NS
	*p* value ^c^	<0.001	<0.05		NS	NS	
Anxiety	Home	2.22 (2.61)	2.43 (2.65)	NS	3.95 (2.30)	3.55 (2.50)	<0.05
	Hospice	2.47 (2.80)	2.04 (2.53)	NS	4.05 (2.53)	2.71 (2.74)	<0.001
	Hospital	2.32 (2.52)	1.34 (1.57)	<0.05	4.08 (1.98)	1.64 (1.66)	<0.001
	*p* value ^b^	NS	<0.05		NS	<0.001	
Breathlessness	Home	1.97 (2.65)	1.64 (2.50)	<0.001	4.02 (2.48)	3.15 (2.72)	<0.001
	Hospice	1.75 (2.69)	1.38 (2.51)	NS	4.47 (2.51)	3.25 (3.17)	<0.05
	Hospital	1.86 (3.01)	1.23 (2.41)	NS	5.47 (2.59)	2.87 (3.42)	<0.05
	*p* value ^c^	NS	NS		NS	NS	
Nausea	Home	1.86 (2.45)	1.59 (2.28)	<0.01	3.70 (2.26)	2.82 (2.46)	<0.001
	Hospice	1.05 (2.02)	0.51 (1.43)	<0.001	3.23 (2.36)	1.17 (1.84)	<0.001
	Hospital	0.66 (1.67)	0.32 (0.83)	NS	3.62 (2.20)	0.62 (1.41)	<0.05
	*p* value ^c^	<0.001	<0.001		NS	NS	

NS, not statistically significant; SD, standard deviation. ^a^ Measured using a numeric rating scale from 0 to 10; ^b^
*p* value for difference between day 7 and day 0; ^c^
*p* value for difference between care settings.

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
