# Peer review of "Impact of Palliative Care in Evaluating and Relieving Symptoms in Patients with Advanced Cancer. Results from the DEMETRA Study"

_ijerph, 2020, doi:10.3390/ijerph17228429_

Round 1

Reviewer 1 Report

The authros have presented a descriptive study which is a sub-study of a larger trial that is presented elsewhere. They have duly acknowledged this fact. I have a few recommendations to improve the presentation of this study and some questions.

  1. Table 4 and 5 have P-values listed, however, it is not clear what is the significance of this P-value. I think it would be better if only changes in intensity of symptoms are presented rather than the number of patients. For example in the tab under asthenia, the number of patients do not change, but the intensity changes and the P-value is significant only for intensity (which makes sense, as same number of people may feel tired even at 7 days, but may feel less tired).

2. It is not clear when the palliative care was introduced to the patient- was it towards the end of treatment, signing of hospice, or whether it was at the time of diagnosis. This needs to be clarified further. 

3. It will be nice to know if these findings can be correlated with death. Perhaps, a cohort can be created where patients received palliative care while receiving treatment versus those who received palliative care after completing treatment. It would be interesting to know if palliative care addition made a difference in Quality of life, intensity of symptoms and even longetivity of life. 

Reviewer 2 Report

I appreciate the opportunity to review the paper. The paper evaluated the effect of palliative care in patients with terminal cancer. The description was well written and sounded in line with the results itself. However, I am afraid that the study design itself is not reasonable. For instance, the authors mentioned that the stages of the participants were terminal by mentioning that the survival days when introduced to palliative care was 29 days in median, they still lack objective scores, such as palliative scale. These need to be mentioned in the profile of the participants. Moreover, the authors mention that there were siginificant decrease in intensity of main symptoms, it is still questionable how clinically significant it is to reduce 1-point in score. There are other questions such as 1)why the participants were introduced to palliative care, which I suspect it varies among patients, 2) why were they not introduce to palliative care earlier. These need to be elucidated.   

Reviewer 3 Report

Thank you for the opportunity of reviewing this article. It is a very interesting research and it is well-written article. Some minor revisions:

  1. According to the first paragraph of discussion, missing data of patients after 7 days was by death. It would be desirable to describe that when the figure 1 is described. In this line, it is suggested to describe the figure 1 in the point 2.1 of the methodology (better than in the statistical analysis). This subsection could be called “Study design and participants”.
  2. The analysis of changes of symptons at 7 days only was possible in some patients, those alive patients. It could led to selection bias and overestimation of results. This point should be discussed as potential limitation in the Discussion section.
  3. It is suggested to be exact when the results are described: 51,7% were female instead of 51%, for example.
  4. It is suggested to include Supplementary Table 4 in the main text because it is one of the main objective of the article. Maybe there are so many Tables. Some information of them could be summarized in text or some tables could be collapsed.
  5. Tables should be self-explained. It would be desirable to include the range of scores in Table 3 and Table 4 to facilitate the interpretation of intensity. In this line, the categorization used for anxiety and depression in the Supplementary Table 4 should be explained in the main text (in the Methodology section).
  6. It is not clear the difference between “all patients” and “symptomatic patients”. Please, explain the difference in more detail.

Round 2

Reviewer 1 Report

My questions have been answered satisfactorily.

Reviewer 2 Report

I have no further comments.